# Sulodexide Increases Glutathione Synthesis and Causes Pro-Reducing Shift in Glutathione-Redox State in HUVECs Exposed to Oxygen–Glucose Deprivation: Implication for Protection of Endothelium against Ischemic Injury

**DOI:** 10.3390/molecules27175465

**Published:** 2022-08-25

**Authors:** Klaudia Bontor, Bożena Gabryel

**Affiliations:** Department of Pharmacology, School of Medicine in Katowice, Medical University of Silesia, 40-752 Katowice, Poland

**Keywords:** sulodexide, endothelial cells, ischemia, apoptosis, oxidative stress, GSH, GSSG, GCLc, GSS, redox potential

## Abstract

Sulodexide (SDX), a purified glycosaminoglycan mixture used to treat vascular diseases, has been reported to exert endothelial protective effects against ischemic injury. However, the mechanisms underlying these effects remain to be fully elucidated. The emerging evidence indicated that a relatively high intracellular concentration of reduced glutathione (GSH) and a maintenance of the redox environment participate in the endothelial cell survival during ischemia. Therefore, the aim of the present study was to examine the hypothesis that SDX alleviates oxygen–glucose deprivation (OGD)-induced human umbilical endothelial cells’ (HUVECs) injury, which serves as the in vitro model of ischemia, by affecting the redox state of the GSH: glutathione disulfide (GSSG) pool. The cellular GSH, GSSG and total glutathione (tGSH) concentrations were measured by colorimetric method and the redox potential (ΔEh) of the GSSG/2GSH couple was calculated, using the Nernst equation. Furthermore, the levels of the glutamate–cysteine ligase catalytic subunit (GCLc) and the glutathione synthetase (GSS) proteins, a key enzyme for de novo GSH synthesis, were determined using enzyme-linked immunoassay (ELISA). We demonstrated that the SDX treatment in OGD conditions significantly elevated the intracellular GSH, enhanced the GSH:GSSG ratio, shifting the redox potential to a more pro-reducing status. Furthermore, SDX increased the levels of both GCLc and GSS. The results show that SDX protects the human endothelial cells against ischemic stress by affecting the GSH levels and cellular redox state. These changes suggest that the reduction in the ischemia-induced vascular endothelial cell injury through repressing apoptosis and oxidative stress associated with SDX treatment may be due to an increase in GSH synthesis and modulation of the GSH redox system.

## 1. Introduction

Ischemia (oxygen and nutrient deficiency) plays a crucial role in the pathogenesis of cardiovascular diseases, including myocardial ischemia, ischemic stroke, and chronic or acute limb ischemia [1,2]. The vascular endothelium is the first site of ischemic injury. Multiple mechanisms can lead dysfunction already during ischemia to endothelial dysfunction and damage, e.g., the depletion of energy stores, disturbances in protein synthesis, increase in pro-inflammatory mediators, induction of adhesion molecules, and the modulation of ion channels and gap junction proteins [3].

One of the undesired consequences of ischemia is an increased potential for oxidative damage of cellular macromolecules, such as DNA, proteins, and lipids by reactive oxygen species (ROS). The major sources of intracellular ROS are mitochondrial electron transport, arachidonic acid pathway and activities of cellular oxidases [4]. The oxidative stress generating from the excessive ROS production and decreased availability of antioxidants, such as glutathione, contributes to the loss of endothelial barrier function, glycocalyx degradation, apoptosis, and vascular injury [5,6]. Thus, a possible therapeutic target for endothelial protection is the enhancement of the endogenous antioxidant defense system.

Glutathione is a tripeptide (γ-L-glutamyl-L-cysteinyl-glycine) that serves as a major endogenous non-enzymatic antioxidant against the oxidative stress and endothelial dysfunction caused by ischemia [7]. Moreover, glutathione deficiency is known to contribute to the apoptosis of ischemic endothelial cells [8]. Glutathione exists in two forms, reduced (GSH) and oxidized (GSSG). GSH is involved in the neutralization of ROS, being reduced by oxidation to GSSG. The GSH:GSSG ratio is a critical determinant of the cellular redox status [9]. Meanwhile, the redox potential (ΔEh) of the GSSG/2GSH couple, quantitatively calculated according to the Nernst equation, is considered as a major redox buffer in the cells. Furthermore, it is currently believed that the redox state of the GSSG/2GSH couple may an indicator of the biological state of the cells, including apoptosis or necrosis [10].

GSH is synthesized de novo from precursor amino acids (glutamate, cysteine, and glycine) in a two-step ATP-dependent enzymatic process catalyzed by glutamate–cysteine ligase (GCL) and glutathione synthase (GSS). GCL catalyzes the rate-limiting reaction between glutamate and cysteine, whose product is γ-glutamylcysteine (γGC). Then, GSS links γGC to glycine to form GSH molecule [11]. GCL consists of catalytic (GCLc) and modifier (GCLm) subunits. GCLc exhibits all of the catalytic activity of the enzyme, whereas GCLm is enzymatically inactive but regulates the binding activity of GCLc to its substrates [12].

GSH is also synthesized via the salvage pathway, which either involves the reduction in GSSG by glutathione reductase or uses precursors formed from the hydrolysis of GSH or its conjugates by γ-glutamyltranspeptidase at the surface of the cell membrane transported back into cells as amino acids or dipeptides [13].

Decreased GSH levels and GSH:GSSG ratio were reported in many ischemic vascular diseases and may reflect a common pathophysiological mechanism [7]. Therefore, the search for suitable therapeutic agents to improve the GSH content and GSH–redox balance could be an effective means of treating ischemic injury.

SDX is a mixture of glycosaminoglycans (GAGs), purified from porcine mucosa consisting of 80% low-molecular weight heparin and 20% dermatan sulfate [14]. SDX possesses a wide range of pharmacological properties, such as antithrombotic, profibrinolytic, and lipid-lowering activity [15]. This drug is used in clinical practice for the treatment of chronic venous and arterial diseases [14,16]. Recently, SDX was associated with an improvement in the therapy for venous leg ulcers [17], deep vein thrombosis [18], cerebrovascular disorders [15], proteinuria, and cardiovascular disease in diabetes [17,19].

In vitro and in vivo studies have shown that SDX exerts anti-inflammatory [20], antioxidant [21], immunomodulatory [22], anti-proliferative [23], antiproteolytic [24], and vasculo-protective features [25].

The studies on the endothelial protective effect of SDX obtained the desired results. For example, SDX was reported to maintain or restore the integrity of the glycocalyx, a gel-like layer covering the luminal surface of vascular endothelium, possibly by providing precursors of endothelial GAGs [26]. SDX also alleviated the endothelial dysfunction in streptozotocin-induced diabetes in rats, by reducing the number of the circulating endothelial cells and improving the endothelium-dependent relaxation in the small arteries [27]. Furthermore, SDX has been shown to exert potent anti-senescent and anti-inflammatory effects in both venous [28], and arterial endothelial cells [29].

Several studies have shown that the antioxidant properties of SDX underlie many other bioactivities of this drug [30,31,32,33]. However, only a few studies have investigated the molecular mechanism of SDX against ischemic damage [34,35,36,37,38,39]. We previously described that SDX induces GSH-related genes in ischemic endothelial cells [37], but the direct contribution of GSH to the protective effects of SDX has yet to be established.

In this study, we investigated the ability of SDX to prevent apoptosis and oxidative stress in the human umbilical vein endothelial cells (HUVECs) induced by oxygen–glucose deprivation (OGD), a commonly used model of simulated ischemia in vitro, by affecting the intracellular tGSH, GSH, and GSSG concentrations, as well as the redox state of the GSH:GSSG pool. Furthermore, the levels of the GCLc and GSS proteins, the key enzymes for de novo GSH synthesis, were determined.

## 2. Results

### 2.1. Effect of SDX on Apoptosis

As shown in Figure 1A (upper panel), after 6 h of simulated ischemia, many of the endothelial cells detached from each other and exhibited cytoplasmic shrinkage. However, the SDX treatment in OGD suppressed the morphological changes of the HUVECs, and increased the number of surviving cells. These results were further confirmed by analysis of the apoptotic features, such as changes in nuclear morphology using Hoechst 33342. The Hoechst 33342 fluorochrome is used to detect the compacted state of chromatin. As shown in Figure 1A (middle panel), the control cells exhibited uniformly dispersed chromatin. However, the cells exposed to OGD for 6 h showed the typical features of apoptosis (chromatin condensation, nuclear shrinkage, and apoptotic bodies formation). Moreover, we found that the SDX treatment significantly inhibited the OGD-induced apoptotic nuclear damages.

As quantified in Figure 1B, AI was strongly increased after 6 h of OGD (median 29%, range 15–42) compared to the Control group (median 1.98%, range 1–7.6). A statistically significant decrease in AI was detected when the ischemic HUVECs were treated with SDX (median 13%, range 6–18).

### 2.2. Effect of SDX on Intracellular ROS Accumulation

To analyze whether the antiapoptotic action of SDX is associated with a decrease in ROS level, the HUVECs were incubated with the cell-permeable dye, CellROX Green Reagent, which exhibits green fluorescence and binds to nuclear DNA only upon oxidation. The fluorescence microscopy of the normoxic cells showed a weak punctate green fluorescence pattern, indicating basal mitochondrial ROS production (Figure 1A, lower panel). After 6 h of OGD, the staining pattern changed to bright nuclear fluorescence, showing the oxidation of CellROX Green Reagent and the binding of the oxidized dye to nuclear DNA. The co-incubation of the OGD-treated cells with SDX efficiently suppressed the bright nuclear fluorescence, indicating inhibition of the ROS accumulation.

As shown in Figure 1C, the intensity of the CellROX green fluorescence was dramatically increased in the OGD group (median 64.90, range 48.44–82.70) compared with the Control group (median 24.64, range 13.40–31.03). After treatment with SDX in OGD, the ROS-stimulated oxidation of the CellROX Green Reagent was significantly lower (median 38.44, range 33.45–44) compared to the OGD group.

Importantly, the decreased production of the ROS correlates with a reduction in the endothelial cell apoptosis, with the highest levels observed in the OGD + SDX group (Figure 1D, R^2^ = 0.63 and *p* < 0.001).

### 2.3. Effect of SDX on Intracellular GSH Content, GSH:GSSG Ratio and Redox Potential

To confirm that SDX may exert antioxidant and endothelial protective effects by ameliorating the glutathione-dependent redox imbalance caused by ischemic injury, the concentrations of tGSH, GSH, and GSSG, the GSH:GSSG ratio, and the changes in the redox potential ΔEh of the GSSG/2GSH couple were determined.

Our data indicate that the intracellular tGSH concentrations gradually decreased in response to 1, 3, and 6 h of OGD (median 17.01 nmol/mg protein, range 13.17–18.71; median 10.29 nmol/mg protein, range 8.9–11.61; median 2.6 nmol/mg protein, range 1.14–3.56, respectively) compared to the Control group (median 23.26 nmol/mg protein, range 17.78–25.4).

As shown in Figure 2A, the SDX treatment for 1 and 3 h in OGD significantly elevated the intracellular tGSH (median 29.12 nmol/mg protein, range 26.96–33.64; median 30.26 nmol/mg protein, range 27.44–32.69, respectively). Compared with the Control group, the intracellular tGSH of the HUVECs treated with SDX in OGD for 6 h decreased significantly (median 9.3 nmol/mg protein, range 6.33–9.69). Compared with the 6 h OGD-treated cells, the intracellular tGSH was significantly increased after the SDX treatment.

We further separately determined the effect of SDX on the GSH and GSSG concentrations. The results presented in Figure 2B show a significant gradual decrease in the GSH levels from 21.26 ± 2.69 nmol/mg protein in the control cells to 14.9 ± 2.13, 8.42 ± 0.54, and 2.67 ± 0.04 nmol/mg protein after 1, 3, and 6 h of OGD exposure, respectively. The GSSG levels, which were much lower than the GSH levels at baseline (0.87 ± 0.56 nmol/mg protein), increased to 1.77 ± 0.42 and 1.59 ± 0.8 nmol/mg protein after 1 and 3 h of OGD, respectively; and then decreased to 0.16 ± 0.7 nmol/mg protein after 6 h. It was observed that treatment with SDX for 1 and 3 h in OGD resulted in a significant increase in the intracellular GSH concentrations (26.22 ± 1.63 and 27.72 ± 0.83 nmol/mg protein, respectively). After 6 h treatment with SDX and OGD, the intracellular GSH level was significantly reduced (6.88 ± 0.73 nmol/mg protein) compared with the untreated HUVECs, but remained higher than in the corresponding OGD group. In addition, the treatment of the HUVECs with SDX for 1 and 3 h in OGD significantly increased the GSSG content (3.68 ± 0.56 and 2.54 ± 0.58 nmol/mg protein, respectively). The exposure of the cells to SDX for 6 h did not significantly affect the GSSG level, compared to the Control and corresponding OGD group (Figure 2B). The intracellular glutathione-redox balance was expressed as the GSH:GSSG ratio (Figure 2C). A higher GSH:GSSG ratio could make the endothelial cells more resistant to oxidative stress and compensate for the decrease in the GSH levels [10,40]. The intracellular GSH:GSSG ratio after 1, 3, and 6 h of OGD exposure was found to be significantly lower (5.98 ± 0.29; 4.69 ± 1.24; and 4.54 ± 1.42, respectively) than in the control cells (19.48 ± 3.52). The redox balance shifts toward a more reducing state in the ischemic HUVECs treated with SDX for 3 or 6 h (12.27 ± 1.47; 10.74 ± 1.74, respectively). However, there was no significant difference in the GSH:GSSH ratio between the SDX-treated and non-treated groups after 1 h of OGD (7.78 ± 1.78 vs. 5.98 ± 0.29) due to an increase in the intracellular GSSG level (Figure 2B,C).

Furthermore, the redox potential was determined according to the Nernst equation (Figure 2D) [41]. It was well known that an increase in the ΔEh of the GSSG/2GSH couple can lead to apoptosis, and the antioxidants that decrease the ΔEh have antiapoptotic properties [10,42]. It was shown that ΔEh was increased in the cells exposed to OGD for 1, 3, or 6 h (−188 ± 12; −165 ± 5; and −127 ± 10 mV, respectively) compared to the control (−237 ± 5 mV). Moreover, the treatment of the cells with SDX for 1, 3, and 6 h resulted in a significant decrease in ΔEh (−216 ± 10; −232 ± 5; and −211 ± 19 mV, respectively) compared with the corresponding OGD groups without SDX.

### 2.4. Effect of SDX on GCLc and GSS Protein Levels

We next measured the GCLc and GSS protein levels as a surrogate for their enzymatic activity in the ischemic HUVECs treated with SDX at different time points (Figure 3). Some of the studies have found a correlation between the protein contents and enzyme activity levels of GCL and GSS [43,44].

In this study, we revealed that the GCLc protein levels were significantly increased in the cells exposed to 1, 3, and 6 h of OGD alone (median 20.00 ng/mL, range 18.17–20.38; median 17.4 ng/mL, range 15.15–17.67; median 9.37 ng/mL, range 7.86–10.72, respectively) compared to the Control group (median 3.45 ng/mL, range 2.67–4.95). Moreover, the treatment of cells with SDX for 1 and 3 h in OGD resulted in a dramatic increase in the GCLc protein levels (median 35.76 ng/mL, range 30.24–40.84; median 40.37 ng/mL, range 36.9–42.7, respectively) (Figure 3A).

The treatment of the ischemic HUVECs with or without SDX resulted in similar GSS concentration profiles. The GSS concentrations were found to be increased in cells treated with OGD alone for 1, 3 or 6 h (median 17.32 ng/mL, range 14.98–18.48; median 15.06 ng/mL, range 11.13–16.32; median 8.08 ng/mL, range 6.05–10.48, respectively) compared to control (median 3.27 ng/mL, range 2.97–4.29). The protein levels of GSS were significantly increased after 1 and 3 h of incubation with SDX in OGD (median 33.26 ng/mL, range 24.7–38.12; median 37.34 ng/mL, range 36.37–40.25, respectively) (Figure 3B).

Moreover, GCLc and GSS proteins showed a very high correlation with each other (Figure 3C, R^2^ = 0.97, and *p* < 0.001). This is likely due to the fact that both of the enzymes are coordinately regulated and the experimental conditions that induce the catalytic subunit of GCL also induce the GSS expression [45,46].

## 3. Discussion

The main findings of the present study were as follows: (i) the reduced ROS production in the SDX-treated HUVECs under OGD conditions was strongly correlated with attenuation of apoptosis; (ii) the GSH:GSSG ratios were increased by SDX primarily due to the increase in the GSH concentrations; (iii) the redox potentials (ΔEh) of the GSSG/2GSH couple became more negative during SDX treatment, reflecting a pro-reducing shift; (iv) increases in the intracellular GCLc and GSS levels by SDX appear leading to a rapid de novo GSH synthesis, resulting in a prolonged and maintained antioxidant effect (Figure 4).

The oxidative stress and redox imbalance with low intracellular levels of GSH play a crucial role in the pathophysiology of ischemic vascular diseases [7].

Determining the time-dependent changes of the GSH concentrations and the GSH:GSSG ratios is very important, since GSH maintains the reduced redox state required for endothelial cell survival under ischemic conditions [47]. The decreased GSH levels, or the oxidation state of the GSSG/2GSH redox system measured as a decrease in the GSH:GSSG ratio, or an increase in the reduction potential calculated from the Nernst equation have been associated with apoptosis [10]. A parallel increase in the ROS production was also implicated in the apoptosis of vascular endothelium in response to ischemia [48]. In contrast, an increase in the GSH levels, GSH:GSSG ratio, and the resulting decrease in the redox potential of the GSSG/2GSH, coupled with a decrease in ROS production, were associated with the inhibition of apoptosis [10,40]. Therefore, the antioxidants regulating the glutathione redox status through the positive influence on the GSH amount and the GSH:GSSG ratio are thought to be effective in protecting the endothelium from ischemic damage.

SDX has been found to have distinct endothelial protective properties against ischemia due to its direct and indirect antioxidant actions, but it is not clear whether GSH plays a role in these effects [34,49]. We have previously described that SDX can protect the endothelial cells from ischemic injury via the nuclear factor-erythroid-2-related factor (Nrf2)/antioxidant response element (ARE) signaling pathway. Moreover, we have shown that SDX induces the rapid accumulation of Nrf2 in the nuclei of the OGD-stimulated HUVECs, leading to increased expression of the GSH-related genes [36,37].

In the present study, the SDX-associated decrease in endothelial apoptosis appears to be dependent on antioxidant activity, as suggested by the strong positive correlation between ROS and apoptotic cell death, indicating that a decrease in the ROS production led to a significant reduction in apoptosis (Figure 1). SDX was studied in the context of endothelial cell protection under oxidative stress, and the available data are consistent with our findings. A study by Połubińska et al. [50] showed that SDX was able to protect the HUVECs from oxidative stress induced by serum samples obtained from patients with peripheral vascular disease. In addition, another piece of research showed that SDX protected the human retinal endothelial cells (HRECs) from oxidative damage in an in vitro model of diabetic retinopathy [51]. It has been suggested that the underlying anti-oxidative mechanisms of SDX may involve the reduction in the pro-inflammatory cytokine expression and upregulation of superoxide dismutase activity [14]. Furthermore, SDX has been shown to inhibit apoptosis via a direct inhibition of caspase-3 [35].

This study demonstrated that the SDX treatment significantly increased the tGSH levels in the first 3 h of OGD, as also evidenced by an increase in the GSH and GSSG concentrations in the HUVECs (Figure 2A,B). The intracellular GSH level is maintained by de novo synthesis and the salvage pathways. In our experiment, the induction of GSH synthesis by SDX was not accompanied by a concomitant decrease in the GSSG concentrations (Figure 2B). This confirms that the increase in GSH levels after SDX treatment is mediated by de novo synthesis pathway.

The observed changes in the GSH:GSSG ratio indicate that the SDX treatment is associated with a pro-reducing shift in HUVEC exposed to OGD (Figure 2C), as well as an anti-apoptotic and antioxidant effect (Figure 1). The most proximate cause for these increases in the intracellular GSH:GSSG ratios appears to be the significant increase in GSH concentrations, as SDX also slightly increased the GSSG levels. It is worth highlighting that our results of a basal GSH:GSSG ratio (20:1 in the Control group) are in agreement with the study performed by Shrestha et al. [52] on vena cava endothelial cells (VCECs). Since the venous endothelial cells have a lower GSH:GSSG ratio than the arteries (30:1), they are more sensitive to oxidative stress [52,53].

A temporal relationship between a significant decrease in the cellular GSH and GSH:GSSG ratio and the onset of apoptotic cell death was demonstrated [54]. The evidence suggests that an early loss of the GSH:GSSG balance associated with a more oxidized GSH redox potential precedes the ROS-induced activation of the mitochondrial apoptotic pathway [8]. It was also shown that the disruption of de novo GSH synthesis with L-buthionine (S,R)-sulfoximine (BSO) activates the proapoptotic c-Jun N-terminal kinase (JNK), which plays a key role in the ischemia-induced apoptosis of the vascular endothelial cells [55]. Furthermore, a decreased GSH:GSSG associated with oxidizing conditions is responsible for formation of the S-glutathiolated proteins by a thiol/disulfide exchange mechanism between a SH group of protein and GSSG. The S-glutathiolation of proteins involved in both the receptor-mediated extrinsic and mitochondria-mediated intrinsic pathways of apoptosis is well known [8]. This process is observed under various pathological conditions within the cardiovascular system, including hypoxia/ischemia [56]. Thus, the anti-apoptotic effect of SDX appears to be mediated by significantly increased GSH levels and higher GSH:GSSG ratios.

Additionally, the shift in the GSH:GSSG ratios induced by SDX treatment in OGD appears to alter the redox state of the HUVECs toward a more negative potential (Figure 2D). This suggests that SDX induces a pro-reducing redox potential which may be due to an increased intracellular GSH concentration. Therefore, we next quantified the levels of GCLc and GSS, key enzymes for GSH biosynthesis. The ELISA results show that there are significant differences in the levels of GCLc and GSS between the groups treated without or with SDX after 1 and 3 h of OGD. This also supports our hypothesis that a strong SDX-induced increase in the levels of both the GCLc and GSS proteins may lead to the rapid production of GSH and a pro-reducing shift in the GSSG/2GSH couple, resulting in a prolonged and sustained antioxidant effect (Figure 1 and Figure 3).

However, the increase in the endothelial GCLc and GSS protein levels was also observed in the HUVECs exposed to OGD alone compared to the control (Figure 3). At the same time, OGD alone significantly decreased the intracellular GSH (Figure 2). Both GCLc and GSS are inducible enzymes. The mechanisms underlying the ischemia-associated upregulation of GCLc and GSS proteins are unknown. Other researchers have also reported that oxidative stress or glutathione deficiency upregulates the expression of the proteins involved in GSH synthesis [57,58]. Moreover, the study by Krejsa et al. [59], performed on Jurkat cells, reported that the rapid activation of GCL after the H_2_O_2_ treatment was inversely proportional to the relative intracellular GSH content. Thus, our findings may support the idea that the upregulation of GCLc and GSS is likely an early adaptive response of the endothelial cells to the oxidative stress that occurs during an ischemic insult. Although the antioxidant system shows some compensatory adaptation to the ischemic conditions, the ability of the ischemic HUVECs non-treated with SDX to respond positively to ODG stress is impaired. In addition, the tissue GSH concentrations reflect not only intracellular accumulation, but also its efflux from the cells. The decreased GSH levels in response to ischemia may be associated with a cross-membrane export, mediated by multidrug resistance proteins (MRPs) [60]. Both GSH and GSSG are known to transport substrates for MRPs. The ability of MRPs to actively efflux the endogenous GSH has significant implications for ischemic vascular diseases [61]. It was shown that an increased expression of various MRP isoforms leads to decreased GSH concentrations in the endothelial cells, an alternation of redox status, and accelerated apoptosis [62,63]. Moreover, GSH depletion as such was recognized as a major contributor to the redox balance changes associated with apoptotic cell death [64]. Importantly, the MRPs’ expression at the mRNA and protein levels and their transporter activities were observed in the HUVECs [65]. In view of the above, we cannot exclude the possibility that the endothelial protective mechanisms of SDX may involve some inhibitory effects on the efflux of cellular GSH. This will be the subject of our further research.

In summary, our study shows for the first time that SDX at a clinically relevant concentration (0.5 LRU/mL) and duration of action [20] significantly affects the glutathione levels, increases the GSH:GSSG ratio, and increases the intracellular redox environment rate of reduction under cell-damaging ischemic conditions. Furthermore, the marked and potent induction of GCLc and GSS in response to SDX may confer an increased ability to rapidly synthesize GSH in the ischemic endothelial cells.

## 4. Materials and Methods

### 4.1. Cell Culture

The HUVECs were cultured as previously described [36]. In brief, the HUVEC cell line purchased from Clonetics (Lonza, Verviers, Belgium) was cultured in 75 cm^2^ tissue culture flasks (CytoOne; USA Scientific, Ocala, FL, USA) at 1 × 10^4^ cells/flask in endothelial basal medium (EGM-2; Clonetics) with EGM-2 BulletKit (Clonetics) at 37 °C in a humidified atmosphere of 5% CO_2_ and 95% air. The medium was replaced every 3 days and the cells were subcultured after reaching around 90% confluence. The cells at passages three to five were used in the experiments.

### 4.2. In Vitro Model of Simulated Ischemia

In vitro ischemia was induced in the HUVECs by OGD as previously described [36]. Briefly, the culture medium (EGM-2 with EGM-BulletKit) was removed and the HUVECs were rinsed twice with glucose-free Dulbecco’s Modified Eagle’s Medium (DMEM, no glucose; Thermo Fisher Scientific, Waltham, MA, USA), previously equilibrated in a hypoxic chamber (Galaxy 48 R incubator; Eppendorf/Galaxy Corporation, Enfield, CT, USA), supplemented with 3% O_2_, 5% CO_2_, and 92% N_2_ at 37 °C. For simulated ischemia in vitro, the confluent cells in DMEM without glucose were transferred to a hypoxic chamber flooded with 92% N_2_, 5% CO, and 3% O_2_ and incubated at 37 °C. The control cells in standard EGM-2 supplemented with EGM-2 BulletKit (normoxia) were not treated with OGD.

### 4.3. Drug Administration and Experimental Groups

The SDX (Vessel Due F, 300 LRU/mL) was purchased from Alfasigma S.p.A. (Bologna, Italy). The HUVECs were treated with 0.5 LRU/mL SDX during OGD. The concentration of SDX was based on the literature data [20]. The drug was added directly to the ischemic medium of the OGD, and the samples were harvested immediately after OGD. The cell cultures exposed only to simulated ischemia in vitro (OGD groups) were maintained in a hypoxic chamber for the same time as the cells treated with SDX during OGD.

Firstly, the HUVECs (5 × 10^4^) were plated in 35-mm dishes and treated with or without SDX for 6 h in OGD to determine the effect of SDX on apoptosis and ROS accumulation. The cells were randomly divided into three groups: Control group; OGD group; and OGD + SDX group.

Then, the HUVECs were treated with SDX for 1, 3, or 6 h in OGD to determine the effect of SDX on total GSH (tGSH), GSH, GSSG, GCLc and GSS. For the GCLc and GSS protein assays, the cells were seeded at 1 × 10^4^/well on 24-well plates, and for GSH and GSSG measurement, the cells were cultured in 35-mm dishes at a density of 5 × 10^4^/dish. The cells were randomly assigned to the following experimental groups: Control group; and six groups of cells treated without/with SDX in accordance with OGD exposure time of 1, 3, and 6 h.

### 4.4. Apoptosis Assay

To detect the apoptotic cells, nuclear staining was performed with Hoechst 33342 (Sigma-Aldrich, St. Louis, MO, USA). The cells were rinsed with phosphate-buffered saline (PBS) and fixed with a 4% formaldehyde (in PBS) for 20 min at room temperature. After fixation, the cells were washed twice with PBS and stained with Hoechst 33342 (5 μg/mL) for 5 min in dark. The stained cells were washed three times with PBS and examined with a fluorescent microscope (Nikon TS-100 F, Nikon, Tokyo, Japan). The apoptotic cells were distinguished from the viable cells by condensed chromatin and shrunken nuclei, and by the higher intensity of blue fluorescence of the nuclei. The data were presented as apoptotic index (AI) defined as the percentage of apoptotic cells, according to the equation:AI (%) = (number of apoptotic cells/ total number of cells) × 100(1)

The images were acquired from 12 randomly selected fields from three culture dishes in each group. The post-image acquisition analysis was performed using ImageJ software (1.48v, NIH, Bethesda, MD, USA, http://imagej.nih.gov.ij/).

### 4.5. Detection of Intracellular ROS

The intracellular ROS were analyzed by using the fluorogenic CellROX^®^ Green Reagent (Life Technologies, Molecular Probes, Eugene, OR, USA), according to the manufacturer’s instructions. After treatment, CellROX^®^ Green Reagent (5 μM) was added to the cells and incubated at 37 °C for 30 min. The CellROX^®^ then was removed and the cells were rinsed three times with PBS. The cells were fixed in 4% formaldehyde (in PBS) for 20 min in the dark before detection. The microscopic images of 12 randomly selected fields from three culture dishes in each group were acquired using the Nikon TS-100 F fluorescence microscope equipped with a Nikon DS Ri1-U2 camera and NIS-BR imaging software version 4.6.0 (Nikon, Tokyo, Japan). The intensities of the fluorescent signals were analyzed and quantified by using ImageJ software (1.48v, NIH, USA; http://imagej.nih.gov.ij/).

### 4.6. Measurement of GSH

The total, reduced, and oxidized GSH levels were determined using a Glutathione Colorimetric Detection Kit (Invitrogen^TM^, Life Technologies Co., Frederick, MD, USA), according to the manufacturer’s instructions. Briefly, after treatment with/without SDX for 1, 3, or 6 h in OGD, the HUVECs were washed once with PBS and immediately precipitated in ice-cold 5% (*w*/*v*) 5-sulfo-salicylic acid (SSA, Sigma-Aldrich, St. Louis, MO, USA). The cells were then scraped and transferred to microcentrifuge tubes. The cell extracts were centrifuged at 14,000 rpm for 10 min at 4 °C. The supernatants were used for subsequent determination of total GSH and GSSG, and the remaining pellets were dissolved in RIPA lysis buffer (Sigma-Aldrich, St. Louis, MO, USA), plus 0.1 M NaOH for the determination of the total protein content.

To measure GSSG, the samples were treated with 2-vinylpirydine (2VP, Sigma-Aldrich, St. Louis, MO, USA). The analysis was performed according to the assay kit’s instructions, and the absorbance was read at 405 nm in a microplate reader (Multiskan Ascent, Labsystems, Helsinki, Finland). The total GSH (tGSH) and GSSG concentrations were determined using the standard curve specific to each run. The amount of reduced GSH was obtained by subtracting GSSG from tGSH. The results were normalized to the total protein concentrations in the samples and expressed as nmol/mg protein. The ratio of GSH:GSSG was used to monitor the intracellular glutathione–redox balance [41].

### 4.7. Redox Potential Calculations

The redox potential (ΔEh) of the GSSG/2GSH couple in the HUVECs was calculated from the GSH and GSSG concentrations, using the Nernst equation:ΔEh = E0 + RT/nF ln [GSSG]/[GSH]^2^(2)
where E0 is the standard potential for the redox couple; R is the gas constant; T is the absolute temperature; *n* is the number of electron transferred (*n* = 2); and F is Faraday’s constant. The E0 value for the GSH/GSSG couple at pH = 7.4 is −264 mV [41].

### 4.8. Determination of Protein Content

The protein content was determined using Bradford Reagent (Sigma-Aldrich, St. Louis, MO, USA) with bovine serum albumin (≥98%) (Sigma-Aldrich, St. Louis, MO, USA) as standard. The absorbance was measured at 595 nm using a microplate reader (Multiskan Ascent, Labsystems, Helsinki, Finland).

### 4.9. Enzyme Linked Immunosorbent Assay (ELISA)

The quantitative determinations of the GCLc and GSS proteins were performed in cell culture lysates, using the Human GCLC (glutamate–cysteine ligase catalytic subunit) ELISA Kit and the Human GSS (Glutathione synthetase) ELISA Kit (Wuhan Fine Biological Technology Co., Ltd., Wuhan, China), according to the manufacturer’s instructions. The optical density (O.D.) values at 450 nm were read in a microplate reader (Multiskan Ascent; Labsystems, Helsinki, Finland). The GCLc and GSS protein concentrations in ng/mL were normalized to the total lysate protein to account for the differences in cell numbers.

### 4.10. Statistical Analysis

All of the statistical analyzes were performed using R version 4.2.1 software (https://www.r-project.org/). The normality and homogeneity of variance were tested with Shapiro–Wilk and Levene tests, respectively. The statistically significant differences (*p* < 0.05) were determined by analysis of variance (one-way ANOVA) followed by subsequent Tukey and Games–Howell multiple range tests as post-hoc analysis. The associations between the GCLc and GSS concentration, and between the apoptotic index and the CellRox fluorescence intensity were evaluated using Pearson’s r correlation coefficients.

## Figures and Tables

**Figure 1 molecules-27-05465-f001:**
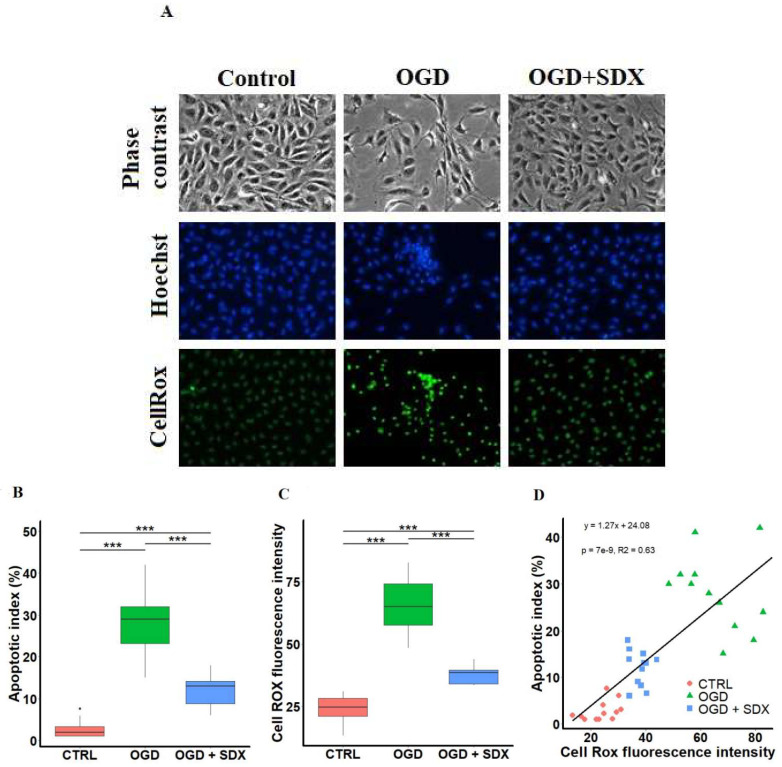
SDX cytoprotective effects against oxidative damage induced by OGD in human endothelial cells. HUVECs were treated for 6 h in OGD in the absence or presence of SDX (0.5 LRU/mL). (**A**) Microscopic observations. Morphology was visualized and photographed under an inverted phase contrast microscope (original magnification ×200). Apoptotic cells were identified by Hoechst 33342 staining and intracellular ROS production was observed using CellROX Green Reagent. The images were examined under a fluorescence microscope (original magnification ×200). Control: normal conditions; OGD: cells exposed to simulated ischemia in vitro only; OGD + SDX: cells exposed to simulated ischemia in vitro and treated with SDX. Apoptotic index (**B**) and quantification of CellROX green fluorescence intensity (**C**) in each corresponding group (*n* = 12). Data in panels (**B**,**C**) are box-plots representing the median and quartiles with the upper and lower limits. Significant results are marked with asterisks (*** *p* < 0.001); (**D**) Correlation between apoptotic index and ROS production. Apoptotic index and ROS generation were determined as described in panel (**A**). Pearson’s correlation coefficient R^2^ = 0.63 was calculated from the linear regression analysis between apoptotic index and CellROX green fluorescence intensity. Control group. [·] is the oulier.

**Figure 2 molecules-27-05465-f002:**
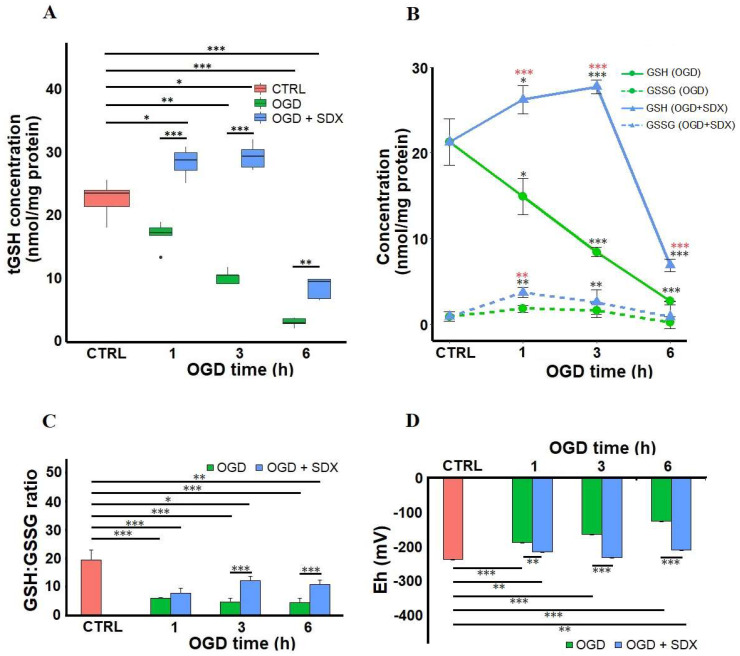
Time-dependent effects of SDX on the intracellular GSH level and redox state in human endothelial cells subjected to OGD. HUVECs were treated for 1, 3, or 6 h in OGD in the absence or presence of SDX (0.5 LRU/mL). The intracellular concentrations of total glutathione (tGSH) (**A**), reduced glutathione (GSH), and oxidized glutathione (GSSG) (**B**) were measured by colorimetric assay (*n* = 4–6). The tGSH, GSH, and GSSG levels were normalized to total protein concentrations and expressed as nmol/mg protein. The GSH:GSSH ratio (**C**) and the GSH redox potential (ΔEh) (**D**) for each incubation period were calculated. CTRL: normal conditions; OGD: cells exposed to simulated ischemia in vitro only; OGD + SDX: cells exposed to simulated ischemia in vitro and treated with SDX. Data in panel (**A**) are box-plots representing the median and quartiles with the upper and lower limits. Each point in panel (**B**) and bar graphs in panels (**C**,**D**) represent mean ± standard deviation (SD). In panels (**A**–**D**), the asterisks indicate the statistically significant differences (* *p* < 0.05; ** *p* < 0.01; *** *p* < 0.001). In panel B, statistically significant differences between OGD groups and OGD + SDX groups are marked with red asterisks (*), while black asterisks (*) are used to mark the differences between the test groups and Control group. [·] is the oulier.

**Figure 3 molecules-27-05465-f003:**
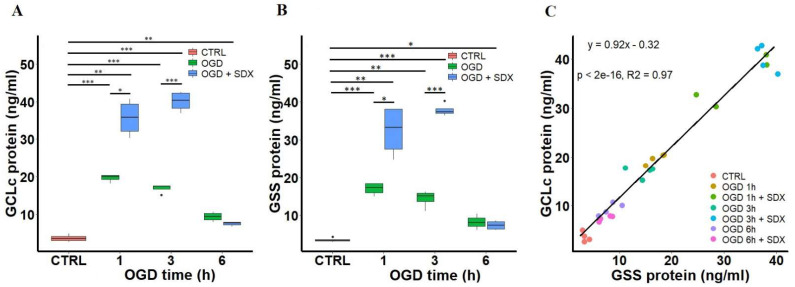
Time-dependent effects of SDX on the protein levels of GCLc and GSS in human endothelial cells subjected to OGD. HUVECs were treated with OGD for 1, 3, or 6 h in the absence or presence of SDX (0.5 LRU/mL). Quantitative analysis of GCLc (**A**) and GSS (**B**) levels in cell lysates was performed by ELISA (*n* = 4). The GCLc and GSS protein concentrations (ng/mL) were normalized to total lysate protein. CTRL: normal conditions; OGD: cells exposed only to simulated ischemia in vitro; OGD + SDX: cells exposed to simulated ischemia in vitro and treated with SDX. Data in panels (**A**,**B**) are box-plots representing the median and quartiles with the upper and lower limits. Significant results are marked with asterisks (* *p* < 0.05; ** *p* < 0.01; *** *p* < 0.001); (**C**) Correlation between GCLc and GSS protein levels. Pearson’s correlation coefficient R^2^ = 0.97 was calculated from the linear regression analysis between concentrations of GCLc and GSS in cell lysates obtained from ELISA. [·] is the oulier.

**Figure 4 molecules-27-05465-f004:**
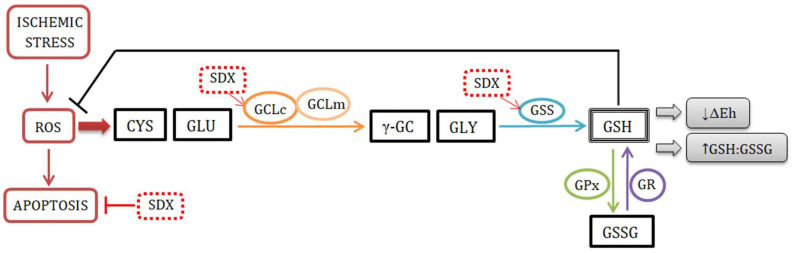
The mechanism of SDX antioxidant action. Abbreviations: ROS, reactive oxygen species; CYS, cysteine; GLU, glutamate; GCLc, catalytic subunit of glutamate–cysteine ligase; GCLm, modifier subunit of glutamate–cysteine ligase; γ-GC, γ-glutamyl cysteine; GLY, glycine; GSS, glutathione synthase; GSH, reduced form of glutathione; GPx, glutathione peroxidase; GSSG, oxidized form of glutathione; GR, glutathione reductase; ∆Eh, redox potential; SDX, sulodexide; →, induction; ┴, inhibition.

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
