# Peer review of "Sulodexide Increases Glutathione Synthesis and Causes Pro-Reducing Shift in Glutathione-Redox State in HUVECs Exposed to Oxygen–Glucose Deprivation: Implication for Protection of Endothelium against Ischemic Injury"

_molecules, 2022, doi:10.3390/molecules27175465_

Round 1
Reviewer 1 Report
The manuscript presented by Klaudia Bontor and Bożena Gabryel shows that Sulodexide protects human endothelial cells from ischemic stress by acting on GSH levels and the cellular redox state. These changes suggest that the reduction of vascular endothelial cell damage is induced by repression of apoptosis and oxidative stress associated with SDX treatment and that this may be due to an increase in GSH synthesis and modulation of the GSH redox system. .
The manuscript is well written, but the conclusions, even if they are original and consistent with the results obtained, are supported by a poor methodology that includes a very low variety of techniques.
Point 1:
To define the cellular apotosis of Huvec cells I ask you to conduct western blot assays and demonstrate an increase in pro-apotic proteins (Caspase family / PARP / mitochondrial proteins from stress caused by ROS) and a decrease in treatment with Sulodexide.
Point 2:
Furthermore, to better define cell apoptosis, it would be useful to visualize the cells in apopotosis differentiating them from the cell population in necresis (annexin V / propidium iodide assay) with the aid of the flow cytometry technique.
Point 3:
There are some formatting errors especially in materials and methods (example line 419: "104 cells / flask in endothelial"); the text must be revised in its entirety.
Step 4:
The references can undoubtedly be updated and integrated with articles from 2021 and also 2022.
Author Response
Response to Reviewer 1 Comments
Point 1: To define the cellular apotosis of Huvec cells I ask you to conduct western blot assays and demonstrate an increase in pro-apotic proteins (Caspase family / PARP / mitochondrial proteins from stress caused by ROS) and a decrease in treatment with Sulodexide.
Response 1: In one of our previous works on Sulodexide, we demonstarted the results of a Western blot analysis, which showed that exposure of HUVECs to OGD resulted in an increase in proapoptotic protein such as cytochrome c in cytosolic fraction and that Sulodexide attenuated an OGD-induced increase in this proapoptotic protein (Gabryel et al., 2016). Moreover, we found that the protective effect of SDX is associated with the prevention of MPTP opening. The cytochrome c release and MPTP opening are two phenomena associated with mitochondrial-pathway-mediated apoptosis caused by ROS. Thus, evidence of an attenuation of proapoptotic proteins by Sulodexide made with Western blot method was presented in other publication by this team (Gabryel et al., Microvasc Res. 2016; 103: 26-35.)
Point 2: Furthermore, to better define cell apoptosis, it would be useful to visualize the cells in apopotosis differentiating them from the cell population in necresis (annexin V / propidium iodide assay) with the aid of the flow cytometry technique.
Response 2: In fact, annexin V / propidium iodide assay is used to distinguish cells undergoing apoptosis from necrotic cells. However, no necrosis was observed at the OGD exposure time (6h) selected by us due to the pharmacokinetics of Sulodexide after intravenous administration. The lack of necrosis was indicated by both morphological observations (Fig. 1) and staining with Live/Dead Kit containing calcein/AM and ethidium homodimer (EthD-1) performed during the preliminary observations (data not shown). According to our own experiences, necrotic endothelial cells can appear in this experimental model after 12-24 hours of exposure to OGD.
Point 3: There are some formatting errors especially in materials and methods (example line 419: "104 cells / flask in endothelial"); the text must be revised in its entirety.
Response 3: The text has been revised for formatting errors.
Point 4: The references can undoubtedly be updated and integrated with articles from 2021 and also 2022.
Response 4: Several articles have been updated.
Please see the attachment.

Reviewer 2 Report
1. The title is too long.
2. The rationale for proposed link between SDX and GSH needs to be stronger.
3. The introduction needs to be more focused on what was examined in this study. And remove the last paragraph.
4. It is not readily understood what “injury” is being referred to in the hypothesis statement. Is this profibrotic signaling? Does apoptosis mediate fibrosis?
5. How are HUVECs being used to model end-organ damage – is this a disease related to pregnancy? It might be more relevant to use liver or kidney cells.
6. DNA damage and timing to apoptosis should be examined.
7. It is not clear from the figures whether it was determined the relative contribution of SDX-mediated control of GClc and GClm vs. GSS on hypoxia/glucose deprivation-induced apoptosis.
8. The summary diagram in figure 4: If this is still a “potential” mechanism, then this study is incomplete. Furthermore, ischemia per se was not studied here.
Author Response
Response to Reviewer 2 Comments
Point 1: The title is too long.
Response 1: We suggest leaving this title because it reflects the essence of our research. We do hope that this title will be accepted.
Point 2: The rationale for proposed link between SDX and GSH needs to be stronger.
Response 2: Thank you for your suggestion on the stronger emphasis on the relationship between SDX and GSH and we pointed it out (see revised version)
Point 3: The introduction needs to be more focused on what was examined in this study. And remove the last paragraph.
Response 3: What was studied is described in more detail. The last paragraph was removed.
Point 4: It is not readily understood what “injury” is being referred to in the hypothesis statement. Is this profibrotic signaling? Does apoptosis mediate fibrosis?
Response 4: This sentence has been reworded.
Point 5: How are HUVECs being used to model end-organ damage – is this a disease related to pregnancy? It might be more relevant to use liver or kidney cells.
Response 5: HUVECs are one of the most commonly used model systems to study vascular biology and pharmacology in vitro. They have been extensively used as a primary, non-immortalized cell system largely due to the ease by which they are isolated and cultured.
Point 6: DNA damage and timing to apoptosis should be examined.
Response 6: In this study, we have used Hoechst 33342 to detect of condensed DNA and fragmented nuclei. We have not investigated the time-dependent effect of SDX on apoptosis because the late phase of this process after nuclear condensation may start from the fourth hour.
Point 7: It is not clear from the figures whether it was determined the relative contribution of SDX-mediated control of GClc and GClm vs. GSS on hypoxia/glucose deprivation-induced apoptosis.
Response 7: This relationship has not been investigated, since for the reasons mentioned above, time-dependent changes in apoptosis have not been studied.
Point 8: The summary diagram in figure 4: If this is still a “potential” mechanism, then this study is incomplete. Furthermore, ischemia per se was not studied here.
Response 6: The word „potential” has been removed.
We hope that the correction will meet your approval.
Please see the attachment.

Round 2
Reviewer 1 Report
Now it is OK.
Reviewer 2 Report
No further comments.